# EFFICIENT SAMPLING FOR GENERATIVE ADVERSARIAL NETWORKS WITH REPARAMETERIZED MARKOV CHAINS

## ABSTRACT

Recently, sampling methods have been successfully applied to enhance the sample quality of Generative Adversarial Networks (GANs). However, in practice, they typically have poor sample efficiency because of the independent proposal sampling from the generator. In this work, we propose REP-GAN, a novel sampling method that allows general dependent proposals by REParameterizing the Markov chains into the latent space of the generator. Theoretically, we show that our reparameterized proposal admits a closed-form Metropolis-Hastings acceptance ratio. Empirically, extensive experiments on synthetic and real datasets demonstrate that our REP-GAN largely improves the sample efficiency and obtains better sample quality simultaneously.

## 1 INTRODUCTION

Generative Adversarial Networks (GANs) (Goodfellow et al., 2014) have achieved a great success on generating realistic images in recent years (Karras et al., 2019; Brock et al., 2019). Unlike previous models that explicitly parameterize the data distribution, GANs rely on an alternative optimization between a generator and a discriminator to learn the data distribution implicitly. However, in practice, samples generated by GANs still suffer from problems such as mode collapse and bad artifacts.

Recently, sampling methods have shown promising results on enhancing the sample quality of GANs by making use of the information in the discriminator. In the alternative training scheme of GANs, the generator only performs a few updates for the inner loop and has not fully utilized the density ratio information estimated by the discriminator. Thus, after GAN training, the sampling methods propose to further utilize this information to bridge the gap between the generative distribution and the data distribution in a fine-grained manner. For example, DRS (Azadi et al., 2019) applies rejection sampling, and MH-GAN (Turner et al., 2019) adopts Markov chain Monte Carlo (MCMC) sampling for the improved sample quality of GANs. Nevertheless, these methods still suffer a lot from the sample efficiency problem. For example, as will be shown in Section 5, MH-GAN's average acceptance ratio on CIFAR10 can be lower than 5%, which makes the Markov chains slow to mix. As MH-GAN adopts an *independent* proposal $q$, i.e., $q(\mathbf{x}'|\mathbf{x}) = q(\mathbf{x}')$, the difference between samples can be so large that the proposal gets rejected easily.

To address this limitation, we propose to generalize the independent proposal to a general *dependent* proposal $q(\mathbf{x}'|\mathbf{x})$. To the end, the proposed sample can be a refinement of the previous one, which leads to a higher acceptance ratio and better sample quality. We can also balance between the exploration and exploitation of the Markov chains by tuning the step size. However, it is hard to design a proper dependent proposal in the high dimensional sample space $\mathcal{X}$ because the energy landscape could be very complex (Neal et al., 2010).

Nevertheless, we notice that the generative distribution $p_g(\mathbf{x})$ of GANs is implicitly defined as the push-forward of the latent prior distribution $p_0(\mathbf{z})$, and designing proposals in the low dimensional latent space is generally much easier. Hence, GAN's latent variable structure motivates us to design a *structured dependent proposal* with two pairing Markov chains, one in the sample space $\mathcal{X}$ and the other in the latent space $\mathcal{Z}$. As shown in Figure 1, given the current pairing samples $(\mathbf{z}_k, \mathbf{x}_k)$, we draw the next proposal $\mathbf{x}'$ in a bottom-to-up way: 1) drawing a latent proposal $\mathbf{z}'$ following $q(\mathbf{z}'|\mathbf{z}_k)$; 2) pushing it forward through the generator and getting the sample proposal $\mathbf{x}' = G(\mathbf{z}')$; 3)

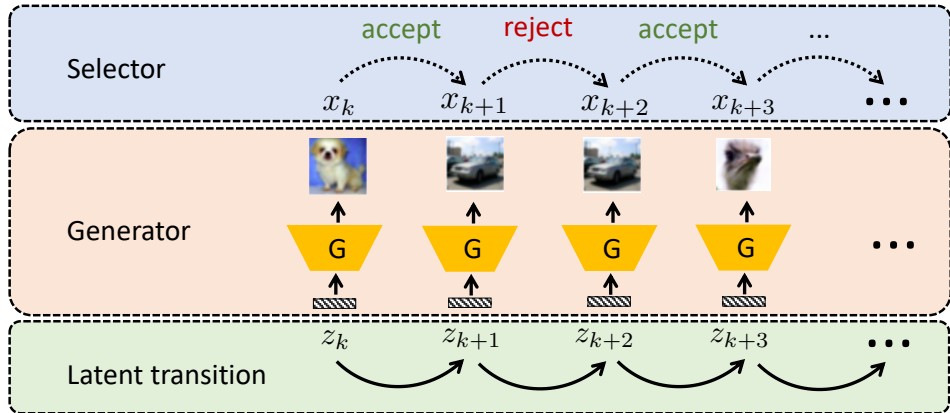

Figure 1: Illustration of REP-GAN's reparameterized dependent proposal with two pairing Markov chains, one in the latent space $\mathcal{Z}$, and the other in the sample space $\mathcal{X}$.

assigning $\mathbf{x}_{k+1} = \mathbf{x}'$ if the proposal $\mathbf{x}'$ is accepted, otherwise rejected $\mathbf{x}_{k+1} = \mathbf{x}_k$. By utilizing the underlying structure of GANs, the proposed reparameterized sampler becomes more efficient in the low-dimensional latent space. We summarize our main contributions as follows:

- We propose a structured dependent proposal of GANs, which reparameterizes the sample-level transition $\mathbf{x} \rightarrow \mathbf{x}'$ into the latent-level $\mathbf{z} \rightarrow \mathbf{z}'$ with two pairing Markov chains. We prove that our reparameterized proposal admits a tractable acceptance criterion.

- Our proposed method, called REP-GAN, serves as a unified framework for the existing sampling methods of GANs. It provides a better balance between exploration and exploitation by the structured dependent proposal, and also corrects the bias of Markov chains by the acceptance-rejection step.

- Empirical results demonstrate that REP-GAN achieves better image quality and much higher sample efficiency than the state-of-the-art methods on both synthetic and real datasets.

## 2 RELATED WORK

Although GANs are able to synthesize high-quality images, the minimax nature of GANs makes it quite unstable, which usually results in degraded sample quality. A vast literature has been developed to fix the problems of GANs ever since, including novel network modules (Miyato et al., 2018), training mechanism (Metz et al., 2017), and alternative objectives (Arjovsky et al., 2017).

Moreover, there is another line of work using sampling methods to improve the sample quality of GANs. DRS (Azadi et al., 2019) firstly proposes to use rejection sampling. MH-GAN (Turner et al., 2019) instead uses the Metropolis-Hasting (MH) algorithm with an independent proposal. DDLS (Che et al., 2020) and DCD (Song et al., 2020) apply gradient-based proposals by viewing GAN as an energy-based model. Tanaka (2019) proposes a similar gradient-based method named DOT from the perspective of optimal transport.

Table 1: Comparison of sampling methods for GANs in terms of three effective sampling mechanisms.

| Method | Rejection step | Markov chain | Latent gradient proposal |
|---|---|---|---|
| GAN | ✗ | ✗ | ✗ |
| DRS (Azadi et al., 2019) | ✓ | ✗ | ✗ |
| MH-GAN (Turner et al., 2019) | ✓ | ✓ | ✗ |
| DDLS (Che et al., 2020) | ✗ | ✓ | ✓ |
| REP-GAN (ours) | ✓ | ✓ | ✓ |

Different from them, our REP-GAN introduces a structured dependent proposal through latent reparameterization, and includes all three effective sampling mechanisms, the Markov Chain Monte Carlo method, the acceptance-rejection step, and the latent gradient-based proposal, to further improve the sample efficiency. As shown in Table 1, many existing works are special cases of our REP-GAN.

Our method also belongs to the thread of works that combine MCMC and neural networks for better sample quality. Previously, some works combine variational autoencoders (Kingma & Welling, 2014) and MCMC to bridge the amorization gap (Salimans et al., 2015; Hoffman, 2017; Li et al., 2017), while others directly learn a neural proposal function for MCMC (Song et al., 2017; Levy et al., 2018; Wang et al., 2018). Our work instead reparameterizes the high-dimensional sample-level transition into a simpler low-dimensional latent space via the learned generator network.

## 3 BACKGROUND

### 3.1 GAN

GAN models the data distribution $p_d(\mathbf{x})$ implicitly with a generator $G : \mathcal{Z} \to \mathcal{X}$ mapping from a low-dimensional latent space $\mathcal{Z}$ to a high-dimensional sample space $\mathcal{X}$,

$$\mathbf{x} = G(\mathbf{z}), \quad \mathbf{z} \sim p_0(\mathbf{z}), \tag{1}$$

where the sample $\mathbf{x}$ follows the generative distribution $p_g(\mathbf{x})$ and the latent variable $\mathbf{z}$ follows the prior distribution $p_0(\mathbf{z})$, e.g., a standard normal distribution $\mathcal{N}(\mathbf{0}, \mathbf{I})$. In GAN, a discriminator $D : \mathcal{X} \to [0, 1]$ is learned to distinguish samples from $p_d(\mathbf{x})$ and $p_g(\mathbf{x})$ in an adversarial way

$$\min_G \max_D \mathbb{E}_{\mathbf{x} \sim p_d(\mathbf{x})} \log(D(\mathbf{x})) + \mathbb{E}_{\mathbf{z} \sim p_0(\mathbf{z})} \log(1 - D(G(\mathbf{z}))). \tag{2}$$

Goodfellow et al. (2014) point out that an optimal discriminator $D$ implies the density ratio between the data and generative distributions

$$D(\mathbf{x}) = \frac{p_d(\mathbf{x})}{p_d(\mathbf{x}) + p_g(\mathbf{x})} \quad \Rightarrow \quad \frac{p_d(\mathbf{x})}{p_g(\mathbf{x})} = \frac{1}{D(\mathbf{x})^{-1} - 1}. \tag{3}$$

### 3.2 MCMC

Markov Chain Monte Carlo (MCMC) refers to a kind of sampling methods that draw a chain of samples $\mathbf{x}_{1:K} \in \mathcal{X}^K$ from a target distribution $p_t(\mathbf{x})$. We denote the initial distribution as $p_0(\mathbf{x})$ and the proposal distribution as $q(\mathbf{x}'|\mathbf{x}_k)$. With the Metropolis-Hastings (MH) algorithm, we accept the proposal $\mathbf{x}' \sim q(\mathbf{x}'|\mathbf{x}_k)$ with probability

$$\alpha(\mathbf{x}', \mathbf{x}_k) = \min\left(1, \frac{p_t(\mathbf{x}') \, q(\mathbf{x}_k|\mathbf{x}')}{p_t(\mathbf{x}_k) \, q(\mathbf{x}'|\mathbf{x}_k)}\right) \in [0, 1]. \tag{4}$$

If $\mathbf{x}'$ is accepted, $\mathbf{x}_{k+1} = \mathbf{x}'$, otherwise $\mathbf{x}_{k+1} = \mathbf{x}_k$. Under mild assumptions, the Markov chain is guaranteed to converge to $p_t(\mathbf{x})$ as $K \to \infty$. In practice, the sample efficiency of MCMC crucially depends on the proposal distribution to trade off between exploration and exploitation.

## 4 THE PROPOSED REP-GAN

In this section, we first review MH-GAN and point out the limitations. We then propose our structured dependent proposal to overcome these obstacles, and finally discuss its theoretical properties as well as practical implementations.

### 4.1 FROM INDEPENDENT PROPOSAL TO DEPENDENT PROPOSAL

MH-GAN (Turner et al., 2019) first proposes to improve GAN sampling with MCMC. Specifically, given a perfect discriminator $D$ and a descent (but imperfect) generator $G$ after training, they take the data distribution $p_d(\mathbf{x})$ as the target distribution and use the generator distribution $p_g(\mathbf{x})$ as an independent proposal

$$\mathbf{x}' \sim q(\mathbf{x}'|\mathbf{x}_k) = q(\mathbf{x}') = p_g(\mathbf{x}'). \tag{5}$$

With the MH criterion (Eqn. (4)) and the density ratio (Eqn. (3)), we should accept $\mathbf{x}'$ with probability

$$\alpha_{\mathrm{MH}}\left(\mathbf{x}', \mathbf{x}_k\right) = \min\left(1, \frac{p_d\left(\mathbf{x}'\right)q\left(\mathbf{x}_k\right)}{p_d\left(\mathbf{x}_k\right)q\left(\mathbf{x}'\right)}\right) = \min\left(1, \frac{D\left(\mathbf{x}_k\right)^{-1} - 1}{D\left(\mathbf{x}'\right)^{-1} - 1}\right). \tag{6}$$

However, to achieve tractability, MH-GAN adopts an independent proposal $q(\mathbf{x}')$ with poor sample efficiency. As the proposed sample $\mathbf{x}'$ is independent of the current sample $\mathbf{x}_k$, the difference between the two samples can be so large that it results in a very low acceptance probability. Consequently, samples can be trapped in the same place for a long time, leading to a very slow mixing of the chain.

A natural solution is to take a *dependent* proposal $q(\mathbf{x}'|\mathbf{x}_k)$ that will propose a sample $\mathbf{x}'$ close to the current one $\mathbf{x}_k$, which is more likely to be accepted. Nevertheless, the problem of such a dependent proposal is that its MH acceptance criterion

$$\alpha_{\mathrm{DEP}}\left(\mathbf{x}', \mathbf{x}_k\right) = \min\left(1, \frac{p_d\left(\mathbf{x}'\right)q\left(\mathbf{x}_k|\mathbf{x}'\right)}{p_d\left(\mathbf{x}_k\right)q\left(\mathbf{x}'|\mathbf{x}_k\right)}\right), \tag{7}$$

is generally intractable because the data density $p_d(\mathbf{x})$ is unknown. Besides, it is hard to design a proper dependent proposal $q(\mathbf{x}'|\mathbf{x}_k)$ in the high dimensional sample space $\mathcal{X}$ with complex landscape. These obstacles prevent us from adopting a dependent proposal that is more suitable for MCMC.

## 4.2 A Tractable Structured Dependent Proposal with Reparameterized Markov Chains

As discussed above, the major difficulty of a general dependent proposal $q(\mathbf{x}'|\mathbf{x}_k)$ is to compute the MH criterion. We show that it can be made tractable by considering an additional pairing Markov chain in the latent space.

As we know, samples of GANs lie in a low-dimensional manifold induced by the push-forward of the latent variable. Suppose that at the $k$-th step of the Markov chain, we have a GAN sample $\mathbf{x}_k$ with latent $\mathbf{z}_k$. Instead of drawing a sample $\mathbf{x}'$ directly from a sample-level proposal distribution $q(\mathbf{x}'|\mathbf{x}_k)$, we first draw a latent proposal $\mathbf{z}'$ from a dependent latent proposal distribution $q(\mathbf{z}'|\mathbf{z}_k)$. Afterward, we push the latent $\mathbf{z}'$ forward through the generator and get the output $\mathbf{x}'$ as our sample proposal.

As illustrated in Figure 1, our bottom-to-up proposal relies on the transition reparameterization with two pairing Markov chains in the sample space $\mathcal{X}$ and the latent space $\mathcal{Z}$. Hence we call it a REP (reparameterized) proposal. Through a learned generator, we transport the transition $\mathbf{x}_k \rightarrow \mathbf{x}'$ in the high dimensional space $\mathcal{X}$ into the low dimensional space $\mathcal{Z}$, $\mathbf{z}_k \rightarrow \mathbf{z}'$, which enjoys a much better landscape and makes it easier to design proposals in MCMC algorithms. For example, the latent target distribution is nearly standard normal when the generator is nearly perfect. In fact, under mild conditions, the REP proposal distribution $q_{\mathrm{REP}}(\mathbf{x}'|\mathbf{x}_k)$ and the latent proposal distribution $q(\mathbf{z}'|\mathbf{z}_k)$ are tied with the following change of variables (Gemici et al., 2016; Ben-Israel, 1999)

$$\log q_{\mathrm{REP}}(\mathbf{x}'|\mathbf{x}_k) = \log q(\mathbf{x}'|\mathbf{z}_k) = \log q(\mathbf{z}'|\mathbf{z}_k) - \frac{1}{2}\log\det J_{\mathbf{z}'}^{\top}J_{\mathbf{z}'}, \tag{8}$$

where $J_{\mathbf{z}}$ denotes the Jacobian matrix of the push-forward $G$ at $\mathbf{z}$, i.e., $[J_{\mathbf{z}}]_{ij} = \partial\mathbf{x}_i/\partial\mathbf{z}_j, \mathbf{x} = G(\mathbf{z})$.

Nevertheless, it remains unclear whether we can perform the MH test to decide the acceptance of the proposal $\mathbf{x}'$. Note that a general dependent proposal distribution does not meet a tractable MH acceptance criterion (Eqn. (7)). Perhaps surprisingly, it can be shown that with our structured REP proposal, the MH acceptance criterion is tractable for general latent proposals $q(\mathbf{z}'|\mathbf{z}_k)$.

**Theorem 1.** *Consider a Markov chain of GAN samples $\mathbf{x}_{1:K}$ with initial distribution $p_g(\mathbf{x})$. For step $k + 1$, we accept our REP proposal $\mathbf{x}' \sim q_{\mathrm{REP}}(\mathbf{x}'|\mathbf{x}_k)$ with probability*

$$\alpha_{\mathrm{REP}}\left(\mathbf{x}', \mathbf{x}_k\right) = \min\left(1, \frac{p_0(\mathbf{z}')q(\mathbf{z}_k|\mathbf{z}')}{p_0(\mathbf{z}_k)q(\mathbf{z}'|\mathbf{z}_k)} \cdot \frac{D(\mathbf{x}_k)^{-1} - 1}{D(\mathbf{x}')^{-1} - 1}\right), \tag{9}$$

*i.e. let $\mathbf{x}_{k+1} = \mathbf{x}'$ if $\mathbf{x}'$ is accepted and $\mathbf{x}_{k+1} = \mathbf{x}_k$ otherwise. Further assume the chain is irreducible, aperiodic and not transient. Then, according to the Metropolis-Hastings algorithm, the stationary distribution of this Markov chain is the data distribution $p_d(\mathbf{x})$ (Gelman et al., 2013).*

*Proof.* Note that similar to Eqn (8), we also have the change of variables between $p_g(\mathbf{x})$ and $p_0(\mathbf{z})$,

$$\log p_g(\mathbf{x})|_{\mathbf{x}=G(\mathbf{z})} = \log p_0(\mathbf{z}) - \frac{1}{2} \log \det J_{\mathbf{z}}^\top J_{\mathbf{z}}. \tag{10}$$

According to Gelman et al. (2013), the assumptions that the chain is irreducible, aperiodic, and not transient make sure that the chain has a unique stationary distribution, and the MH algorithm ensures that this stationary distribution equals to the target distribution $p_d(\mathbf{x})$. Thus we only need to show that the MH criterion in Eqn. (9) holds. Together with Eqn. (3), (7) and (8), we have

$$
\begin{aligned}
\alpha_{\mathrm{REP}}(\mathbf{x}', \mathbf{x}_k) &= \frac{p_d(\mathbf{x}')\, q(\mathbf{x}_k|\mathbf{x}')}{p_d(\mathbf{x}_k)\, q(\mathbf{x}'|\mathbf{x}_k)} = \frac{p_d(\mathbf{x}') q(\mathbf{z}_k|\mathbf{z}') \left(\det J_{\mathbf{z}_k}^\top J_{\mathbf{z}_k}\right)^{-\frac{1}{2}} p_g(\mathbf{x}_k) p_g(\mathbf{x}')}{p_d(\mathbf{x}_k) q(\mathbf{z}'|\mathbf{z}_k) \left(\det J_{\mathbf{z}'}^\top J_{\mathbf{z}'}\right)^{-\frac{1}{2}} p_g(\mathbf{x}') p_g(\mathbf{x}_k)} \\
&= \frac{q(\mathbf{z}_k|\mathbf{z}') \left(\det J_{\mathbf{z}_k}^\top J_{\mathbf{z}_k}\right)^{-\frac{1}{2}} p_0(\mathbf{z}') \left(\det J_{\mathbf{z}'}^\top J_{\mathbf{z}'}\right)^{-\frac{1}{2}} (D(\mathbf{x}_k)^{-1} - 1)}{q(\mathbf{z}'|\mathbf{z}_k) \left(\det J_{\mathbf{z}'}^\top J_{\mathbf{z}'}\right)^{-\frac{1}{2}} p_0(\mathbf{z}_k) \left(\det J_{\mathbf{z}_k}^\top J_{\mathbf{z}_k}\right)^{-\frac{1}{2}} (D(\mathbf{x}')^{-1} - 1)} \\
&= \frac{p_0(\mathbf{z}') q(\mathbf{z}_k|\mathbf{z}') (D(\mathbf{x}_k)^{-1} - 1)}{p_0(\mathbf{z}_k) q(\mathbf{z}'|\mathbf{z}_k) (D(\mathbf{x}')^{-1} - 1)}.
\end{aligned}
\tag{11}
$$

Hence the proof is completed. $\qquad\square$

The theorem above demonstrates the following favorable properties of our method:

- The discriminator score ratio is the same as $\alpha_{\mathrm{MH}}(\mathbf{x}', \mathbf{x}_k)$, but MH-GAN is restricted to a specific independent proposal. Our method instead works for any latent proposal $q(\mathbf{z}'|\mathbf{z}_k)$. When we take $q(\mathbf{z}'|\mathbf{z}_k) = p_0(\mathbf{z}')$, our method reduces to MH-GAN.

- Compared to $\alpha_{\mathrm{DEP}}(\mathbf{x}', \mathbf{x}_k)$ of a general dependent proposal (Eqn. (7)), the unknown data distributions terms are successfully cancelled in the reparameterized acceptance criterion.

- The reparameterized MH acceptance criterion becomes tractable as it only involves the latent priors, the latent proposal distributions, and the discriminator scores.

Combining the REP proposal $q_{\mathrm{REP}}(\mathbf{x}'|\mathbf{x}_k)$ and its tractable MH criterion $\alpha_{\mathrm{REP}}(\mathbf{x}', \mathbf{x}_k)$, we have developed a novel sampling method for GANs, coined as REP-GAN. See Appendix 1 for a detailed description. Moreover, our method can serve as a general approximate inference technique for Bayesian models by bridging MCMC and GANs. Previous works (Marzouk et al., 2016; Titsias, 2017; Hoffman et al., 2019) also propose to avoid the bad geometry of a complex probability measure by reparameterizing the Markov transitions into a simpler measure. However, these methods are limited to explicit invertible mappings without dimensionality reduction. In our work, we first show that it is also tractable to conduct such model-based reparameterization with implicit models like GANs.

## 4.3 A PRACTICAL IMPLEMENTATION

REP-GAN enables us to utilize the vast literature of existing MCMC algorithms (Neal et al., 2010) to design dependent proposals for GANs. We take Langevin Monte Carlo (LMC) as an example. As an Euler-Maruyama discretization of the Langevin dynamics, LMC updates the Markov chain with

$$\mathbf{x}_{k+1} = \mathbf{x}_k + \frac{\tau}{2} \nabla_{\mathbf{x}} \log p_t(\mathbf{x}_k) + \sqrt{\tau} \cdot \boldsymbol{\varepsilon}, \ \ \boldsymbol{\varepsilon} \sim \mathcal{N}(\mathbf{0}, \mathbf{I}), \tag{12}$$

for a target distribution $p_t(\mathbf{x})$. Compared to MH-GAN, LMC utilizes the gradient information to explore the energy landscape more efficiently. However, if we directly take the (unknown) data distribution $p_d(\mathbf{x})$ as the target distribution $p_t(\mathbf{x})$, LMC does not meet a tractable update rule.

As discussed above, the reparameterization of REP-GAN makes it easier to design transitions in the low-dimensional latent space. Hence, we instead propose to use LMC for the latent Markov chain. We assume that the data distribution also lies in the low-dimensional manifold induced by the generator, i.e., $\mathrm{Supp}\,(p_d) \subset \mathrm{Im}(G)$. This implies that the data distribution $p_d(\mathbf{x})$ also has a pairing distribution in the latent space, denoted as $p_t(\mathbf{z})$. They are tied with the change of variables

$$\log p_d(\mathbf{x})|_{\mathbf{x}=G(\mathbf{z})} = \log p_t(\mathbf{z}) - \frac{1}{2} \log \det J_{\mathbf{z}}^\top J_{\mathbf{z}}, \tag{13}$$

Taking $p_t(\mathbf{z})$ as the (unknown) target distribution of the latent Markov chain, we have the following Latent LMC (L2MC) proposal

$$
\begin{aligned}
\mathbf{z}' &= \mathbf{z}_k + \frac{\tau}{2} \nabla_\mathbf{z} \log p_t(\mathbf{z}_k) + \sqrt{\tau} \cdot \boldsymbol{\varepsilon} \\
&= \mathbf{z}_k + \frac{\tau}{2} \nabla_\mathbf{z} \log \frac{p_t(\mathbf{z}_k) \left( \det J_{\mathbf{z}_k}^\top J_{\mathbf{z}_k} \right)^{-\frac{1}{2}}}{p_0(\mathbf{z}_k) \left( \det J_{\mathbf{z}_k}^\top J_{\mathbf{z}_k} \right)^{-\frac{1}{2}}} + \frac{\tau}{2} \nabla_\mathbf{z} \log p_0(\mathbf{z}_k) + \sqrt{\tau} \cdot \boldsymbol{\varepsilon} \\
&= \mathbf{z} + \frac{\tau}{2} \nabla_\mathbf{z} \log \frac{p_d(\mathbf{x}_k)}{p_g(\mathbf{x}_k)} + \frac{\tau}{2} \nabla_\mathbf{z} \log p_0(\mathbf{z}_k) + \sqrt{\tau} \cdot \boldsymbol{\varepsilon} \\
&= \mathbf{z} - \frac{\tau}{2} \nabla_\mathbf{z} \log(D^{-1}(\mathbf{x}_k) - 1) + \frac{\tau}{2} \nabla_\mathbf{z} \log p_0(\mathbf{z}_k) + \sqrt{\tau} \cdot \boldsymbol{\varepsilon}, \quad \boldsymbol{\varepsilon} \sim \mathcal{N}(\mathbf{0}, \mathbf{I}),
\end{aligned}
\tag{14}
$$

where $\mathbf{x}_k = G(\mathbf{z}_k)$. As we can see, L2MC is made tractable by our structured dependent proposal with pairing Markov chains. DDLS (Che et al., 2020) proposes a similar Langevin proposal by formalizing GANs as an implicit energy-based model, while here we provide a straightforward derivation through reparameterization. Our major difference to DDLS is that REP-GAN also includes a tractable MH correction step (Eqn. (9)), which accounts for the numerical errors introduced by the discretization and ensures that detailed balance holds.

## 4.4 EXTENSION TO WGAN

Our method can also be extended to other kinds of GAN, like Wasserstein GAN (WGAN) (Arjovsky et al., 2017). The WGAN objective is

$$
\min_G \max_D \mathbb{E}_{\mathbf{x} \sim p_d(\mathbf{x})}[D(\mathbf{x})] - \mathbb{E}_{\mathbf{x} \sim p_g(\mathbf{x})}[D(\mathbf{x})],
\tag{15}
$$

where $D : \mathcal{X} \to \mathbb{R}$ is restricted to be a Lipschitz function. Under certain conditions, WGAN also implies an approximate estimation of the density ratio (Che et al., 2020),

$$
D(\mathbf{x}) \approx \log \frac{p_d(\mathbf{x})}{p_g(\mathbf{x})} + \text{const} \quad \Rightarrow \quad \frac{p_d(\mathbf{x})}{p_g(\mathbf{x})} \approx \exp(D(\mathbf{x})) \cdot \text{const}.
\tag{16}
$$

Following the same derivations as in Eqn. (11) and (14), we will have the WGAN version of REP-GAN. Specifically, with $\mathbf{x}_k = G(\mathbf{z}_k)$, the L2MC proposal follows

$$
\mathbf{z}' = \mathbf{z}_k + \frac{\tau}{2} \nabla_\mathbf{z} D(\mathbf{x}_k) + \frac{\tau}{2} \nabla_\mathbf{z} \log p_0(\mathbf{z}_k) + \sqrt{\tau} \cdot \boldsymbol{\varepsilon}, \quad \boldsymbol{\varepsilon} \sim \mathcal{N}(\mathbf{0}, \mathbf{I}),
\tag{17}
$$

and the MH acceptance criterion is

$$
\alpha_{REP-W}(\mathbf{x}', \mathbf{x}_k) = \min \left( 1, \frac{q(\mathbf{z}_k | \mathbf{z}') p_0(\mathbf{z}')}{q(\mathbf{z}' | \mathbf{z}_k) p_0(\mathbf{z}_k)} \cdot \frac{\exp(D(\mathbf{x}'))}{\exp(D(\mathbf{x}_k))} \right).
\tag{18}
$$

## 5 EXPERIMENTS

We show our empirical results both on synthetic and real image datasets.

### 5.1 SYNTHETIC DATA

Following DOT (Tanaka, 2019) and DDLS (Che et al., 2020), we apply REP-GAN to the synthetic Swiss Roll dataset, where data samples lie on a Swiss roll manifold in the two-dimensional space. We construct the dataset by scikit-learn with 100,000 samples, and train a WGAN as in Tanaka (2019), where both the generator and discriminator are fully connected neural networks with leaky ReLU nonlinearities. We optimize the model using the Adam optimizer, with $\alpha = 0.0001, \beta_1 = 0.5, \beta_2 = 0.9$. After training, we draw 1,000 samples with different sampling methods.

As shown in Figure 2, with appropriate step size ($\tau = 0.01$), the gradient-based methods (DDLS and REP-GAN) outperform independent proposals (DRS and MH-GAN) by a large margin, while DDLS is more discontinuous on shape compared to REP-GAN. In DDLS, when the step size becomes too large ($\tau = 0.1, 1$), the numerical error of the Langevin dynamics becomes so large that the chain either collapses or diverges. In contrast, those bad proposals are rejected by the MH correction steps of REP-GAN, which prevents the misbehavior of the Markov chain.

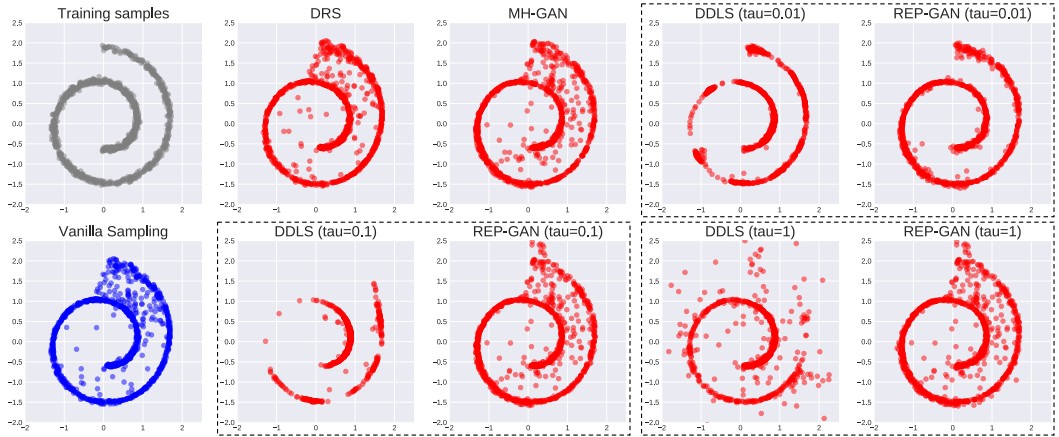

Figure 2: Visualization of samples with different sampling methods on the Swiss Roll dataset. Here tau denotes the Langevin step size in Eqn. (17).

Table 2: Inception Scores (IS) of different sampling methods on CIFAR-10 and CelebA. * For GAN, DRS and MH-GAN, we report the results in Turner et al. (2019). ♯ The results are given by our implementation.

| Method | CIFAR-10 | | CelebA | |
|---|---|---|---|---|
| | DCGAN | WGAN | DCGAN | WGAN |
| GAN* | 2.879 | 3.073 | 2.332 | 2.788 |
| DRS (Azadi et al., 2019)* | 3.073 | 3.137 | 2.869 | 2.861 |
| MH-GAN (Turner et al., 2019)* | 3.379 | 3.305 | **3.106** | 2.889 |
| DDLS (Che et al., 2020)♯ | 3.518 | 3.547 | 2.534 | 2.862 |
| REP-GAN (ours)♯ | **3.851** | **4.035** | 2.686 | **2.943** |

## 5.2 REAL IMAGE DATA

Following MH-GAN (Turner et al., 2019), we conduct experiments on two real-world image datasets, CIFAR-10 and CelebA, for DCGAN (Radford et al., 2015) and WGAN (Arjovsky et al., 2017). Following the conventional evaluation protocol, we initialize each Markov chain with a GAN sample, run it for 640 steps, and take the last sample for evaluation. We collect 50,000 samples to evaluate the Inception Score[1] (Salimans et al., 2016). The step size $\tau$ of our L2MC proposal is $0.01$ on CIFAR-10 and $0.1$ on CelebA. We calibrate the discriminator with Logistic Regression as in Turner et al. (2019).

From Table 2, we can see our method outperforms state-of-the-art methods on both datasets. We also plot the Inception Score and acceptance ratio per epoch in Figure 3 based on our re-implementation. Although the training process of GANs is known to be very unstable, our REP-GAN can still outperform previous sampling methods both consistently (superior in most epochs) and significantly (the improvement is larger than the error bar) as shown in the left panel of Figure 3. In the right panel, we find that the average acceptance ratio of MH-GAN is lower than 0.2 in most cases, while REP-GAN has an acceptance ratio of 0.4-0.8, which is known to be a good tradeoff for MCMC algorithms. We also notice that the acceptance ratio goes down as the training continues. We suspect this is because the distribution landscape becomes complex and a constant sampling step size will produce more distinct samples that are more likely to get rejected.

---

[1]For fair comparison, our training and evaluation follows the the official code of MH-GAN (Turner et al., 2019): https://github.com/uber-research/metropolis-hastings-gans

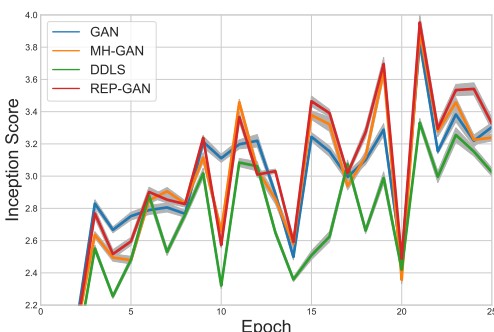 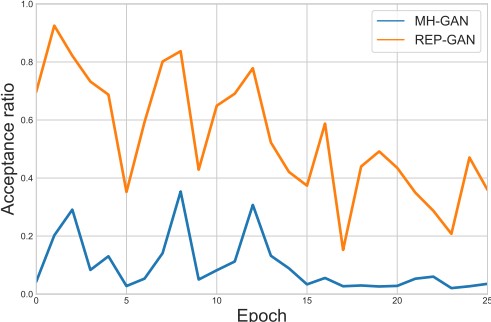

Figure 3: Average Inception Score (left) and acceptance ratio (right) vs. training epoch on CIFAR-10 based on our re-implementation. The standard deviation is shown with shaded error bar (left).

Table 3: Ablation study of our REP-GAN with Inception Scores (IS) and acceptance ratios (Accept) averaged over five adjacent checkpoints. IND refers to the independent proposal of MH-GAN. REP refers to our REP proposal. MH denotes the MH rejection step of the corresponding sampler.

| Proposal | MH | CIFAR10 | | | | CelebA | | | |
| | | DCGAN | | WGAN | | DCGAN | | WGAN | |
| | | Accept | IS | Accept | IS | Accept | IS | Accept | IS |
|---|---|---|---|---|---|---|---|---|---|
| IND | ✗ | - | 3.769 | - | 3.557 | - | 2.431 | - | 2.799 |
| REP | ✗ | - | 3.312 | - | 3.500 | - | 2.465 | - | 2.868 |
| IND | ✓ | 0.033 | 3.735 | 0.084 | 3.509 | **0.678** | **2.500** | 0.747 | 2.879 |
| REP | ✓ | **0.363** | **3.820** | **0.447** | **3.598** | 0.628 | 2.476 | **0.955** | **2.894** |

**Ablation study.** From Table 3, we can see that without the MH correction step, the Langevin steps often result in worse sample quality. Meanwhile, the acceptance is very small on CIFAR-10 without the dependent REP proposal. As a result, our REP-GAN (REP+MH) is the only setup that consistently improves over the baseline and obtains the best Inception Score on each dataset. The only exception is DCGAN on CelebA, where the independent proposal outperforms our REP proposal with a higher acceptance ratio. We believe that this is because the human face samples of CelebA are very alike among each other, such that independent samples from the generator can also be easily accepted. Nevertheless, the acceptance ratio of the independent proposal can be much lower on datasets with diverse sources of images, like CIFAR-10.

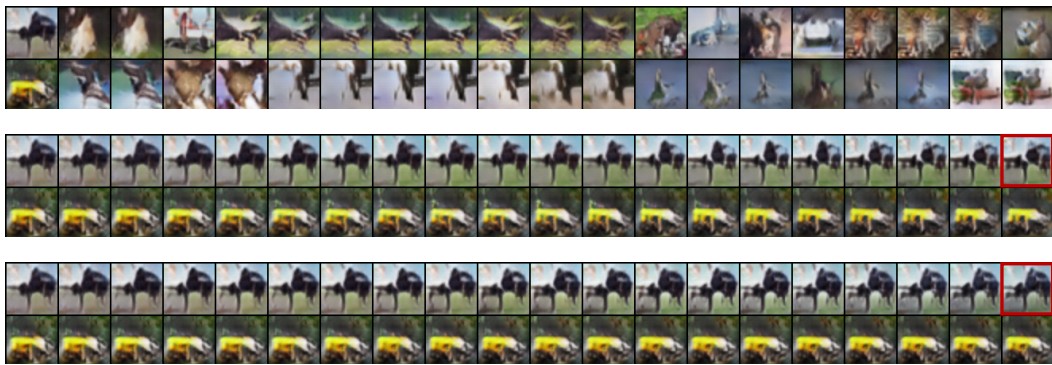

Figure 4: The first 20 steps of two Markov chains with the same initial samples. The chains are generated by MH-GAN (top), DDLS (middle), and REP-GAN (bottom).

**Markov chain visualization.** In Figure 4, we demonstrate two Markov chains sampled with different methods. We can see that MH-GAN is often trapped in the same place because of the independent proposals. DDLS and REP-GAN instead gradually refine the samples with gradient steps. In addition,

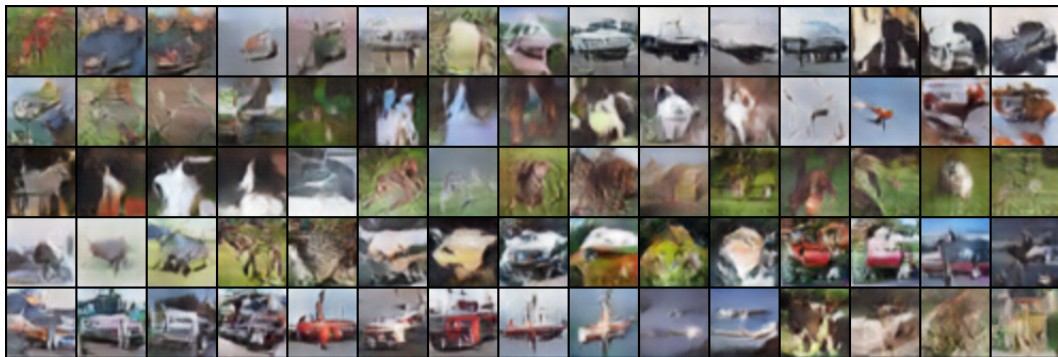

Figure 5: Visualization of 5 Markov chains of our REP proposals (i.e., REP-GAN without the MH rejection steps) with a large step size ($\tau = 1$).

compared the gradient-based methods, we can see that the MH rejection steps of REP-GAN help avoid some bad artifacts in the images. For example, in the camel-like images marked in red, the body of the camel is separated in the sample of DDLS (middle) while it is not in the sample of REP-GAN (bottom). Note that, the evaluation protocol only needs the last step of the chain, thus we prefer a small step size that finetunes the initial samples for better sample quality. As shown in Figure 5, our REP proposal can also produce very diverse images with a large step size.

## 6 CONCLUSION

In this paper, we have proposed a novel method, REP-GAN, to improve the sampling of GAN. We devise a structured dependent proposal that reparameterizes the sample-level transition of GAN into the latent-level transition. More importantly, we first prove that this general proposal admits a tractable MH criterion. Experiments show our method does not only improve sample efficiency but also demonstrate state-of-the-art sample quality on benchmark datasets over existing sampling methods.

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

# A  APPENDIX

## A.1  ASSUMPTIONS AND IMPLICATIONS

Note that our method needs a few assumptions on the models for our analysis to hold. Here we state them explicitly and discuss their applicability and potential impacts.

**Assumption 1.** *The generator mapping* $G : \mathbb{R}^n \to \mathbb{R}^m (n < m)$ *is injective, and its Jacobian matrix* $\left[ \frac{\partial G(\mathbf{z})}{\partial \mathbf{z}} \right]$ *of size* $m \times n$, *has full column rank for all* $\mathbf{z} \in \mathbb{R}^n$.

For the change of variables in Eqn. (8) and (10) to hold, according to Ben-Israel (1999), we need the mapping to be injective and its Jaobian should have full column rank. A mild sufficient condition for injectivity is that the generator only contains (non-degenerate) affine layers and injective non-linearities, like LeakyReLU. It is not hard to show that such a condition also implies the full rankness of the Jacobian. In fact, this architecture has already been found to benefit GANs and achieved state-of-the-art results (Tang, 2020). The affine layers here are also likely to be non-degenerate because their weights are randomly initialized and typically will not degenerate in practice during the training of GANs.

**Assumption 2.** *The discriminator* $D$ *offers a perfect estimate the density ratio between the generative distribution* $p_g(\mathbf{x})$ *and the data distribution* $p_d(\mathbf{x})$ *as in Eqn.* (3).

This is a common, critical, but less practical assumption among the existing sampling methods of GANs. It is unlikely to hold exactly in practice, because during the alternative training of GANs, the generator is also changing all the time, and the a few updates of the discriminator cannot fully learn the corresponding density ratio. Nevertheless, we think it can capture a certain extent information of density ratio which explains why the sampling methods can consistently improve over the baseline at each epoch.

From our understanding, the estimated density ratio is enough to push the generator better but not able to bring it up to the data distribution. This could be the reason why the Inception scores obtained by the sampling methods, can improve over the baselines but cannot reach up to that of real data and fully close the gap, even with very long run of the Markov chains.

Hence, there is still much room for improvement. To list a few, one can develop mechanisms that bring more accurate density ratio estimate, or relax the assumptions for the method to hold, or establishing estimation error bounds. Overall, we believe GANs offer an interesting alternative scenario for the development of sampling methods.

## A.2  ALGORITHM PROCEDURE

We give a detailed description of the algorithm procedure of our REP-GAN in Algorithm 1.

---

**Algorithm 1** GAN sampling with Reparameterized Markov chains (REP-GAN)

---

**Input:** trained GAN with (calibrated) discriminator $D$ and generator $G$, Markov chain length $K$, latent prior distribution $p_0(\mathbf{z})$, latent proposal distribution $q(\mathbf{z}'|\mathbf{z}_k)$;
**Output:** an improved GAN sample $\mathbf{x}_K$;
  Draw an initial sample $\mathbf{x}_1$: 1) draw initial latent $\mathbf{z}_1 \sim p_0(\mathbf{z})$ and 2) push forward $\mathbf{x}_1 = G(\mathbf{z}_1)$;
  **for** each step $k \in [1, K-1]$ **do**
    Draw a REP proposal $\mathbf{x}' \sim q_{\text{REP}}(\mathbf{x}'|\mathbf{x}_k)$: 1) draw a latent proposal $\mathbf{z}' \sim q(\mathbf{z}'|\mathbf{z}_k)$, and 2) push forward $\mathbf{x}' = G(\mathbf{z}')$;
    Calculate the MH acceptance criterion $\alpha_{\text{REP}}(\mathbf{x}_k, \mathbf{x}')$ following Eqn. (9);
    Decide the acceptance of $\mathbf{x}'$ with probability $\alpha_{\text{REP}}(\mathbf{x}_k, \mathbf{x}')$;
    **if** $\mathbf{x}'$ is accepted **then**
      Let $\mathbf{x}_{k+1} = \mathbf{x}', \mathbf{z}_{k+1} = \mathbf{z}'$
    **else**
      Let $\mathbf{x}_{k+1} = \mathbf{x}_k, \mathbf{z}_{k+1} = \mathbf{z}_k$
    **end if**
  **end for**

---

Table 4: Fréchet Inception Distance (FID) of different MCMC sampling methods on CIFAR-10 and CelebA based on our re-implementation.

| Method | CIFAR-10 | | CelebA | |
|---|---|---|---|---|
| | DCGAN | WGAN | DCGAN | WGAN |
| GAN | 100.363 | 153.683 | 227.892 | 207.545 |
| MH-GAN (Turner et al., 2019) | 100.167 | 143.426 | **227.233** | 207.143 |
| DDLS (Che et al., 2020) | 145.981 | 193.558 | 269.840 | 232.522 |
| REP-GAN (ours) | **99.798** | **143.322** | 230.748 | **207.053** |

Table 5: Comparison of computation cost (measured in seconds) of gradient-based MCMC sampling methods of GANs. We report the total time to sample a batch of 500 samples with DCGAN on a NVIDIA 1080 Ti GPU. We initialize the chain with GAN samples and run each chain for 640 steps.

| DDLS | REP-GAN (ours) | REP-GAN w/o MH correction |
|---|---|---|
| 88.94s | 88.85s | 87.62s |

### A.3 ADDITIONAL EMPIRICAL RESULTS

Here we list some additional empirical results of our methods.

**Fréchet Inception Distance (FID).** We additionally report the comparison of Fréchet Inception Distance (FID) in Table 4. Because previous works do not report FID on these benchmarks, we report our re-implementation results instead. We can see the ranks are consistent with the Inception scores in Table 2 and our method is superior in most cases.

**Computation overhead.** In Table 5, we compare different gradient-based sampling methods of GANs. Comparing DDLS and our REP-GAN, they take 88.94s and 88.85s, respectively, hence the difference is negligible. Without the MH-step, our method takes 87.62s, meaning the additional MH-step only costs 1.4% computation overhead, which is also negligible, but it brings a significant improvement of sample quality as shown in Table 3.

**Markov chain visualization on CelebA.** We demonstrate two Markov chains on CelebA with different MCMC sampling methods of WGAN in Figure 6. We can see that on CelebA, the acceptance ratio of MH-GAN becomes much higher than that on CIFAR-10. Nevertheless, the sample quality is still relatively low. In comparison, the gradient-based method can gradually refine the samples with Langevin steps, and our REP-GAN can alleviate image artifacts with MH correction steps.

### A.4 MULTI-MODAL EXPERIMENTS

Aside from the manifold learning example shown in Figure 2, we additionally conduct experiments to illustrate the performance of our sampling methods for multi-modal distributions.

**25-Gaussians.** To begin with, we consider the 25-Gaussians dataset widely discussed in previous work (Azadi et al., 2019; Turner et al., 2019; Che et al., 2020). The 25-Gaussians dataset is generated by a mixture of twenty-five two-dimensional isotropic Gaussian distributions with variance 0.01, and means separated by 1, arranged in a grid. We train a small Wasserstein GAN model with the standard WGAN-GP objective following the setup in Tanaka (2019). After training, we draw 1,000 samples with different sampling methods. Similarly, we starts a Markov chain with a GAN sample, run it for 100 steps, and collect the last example for evaluation.

As shown in Figure 7, compared to MH-GAN, the gradient-based methods (DDLS and ours) produce much better samples close to the data distribution with proper step size. Comparing DDLS and ours, DDLS tends to concentrate so much on the mode centers that its standard deviation can be even smaller than that of data distribution. Instead, our method preserves more sample diversity while concentrating on the mode centers. When the step size becomes larger, the difference becomes more obvious. When $\tau = 0.1$, as marked with blue circles, the samples of DDLS become so concentrated

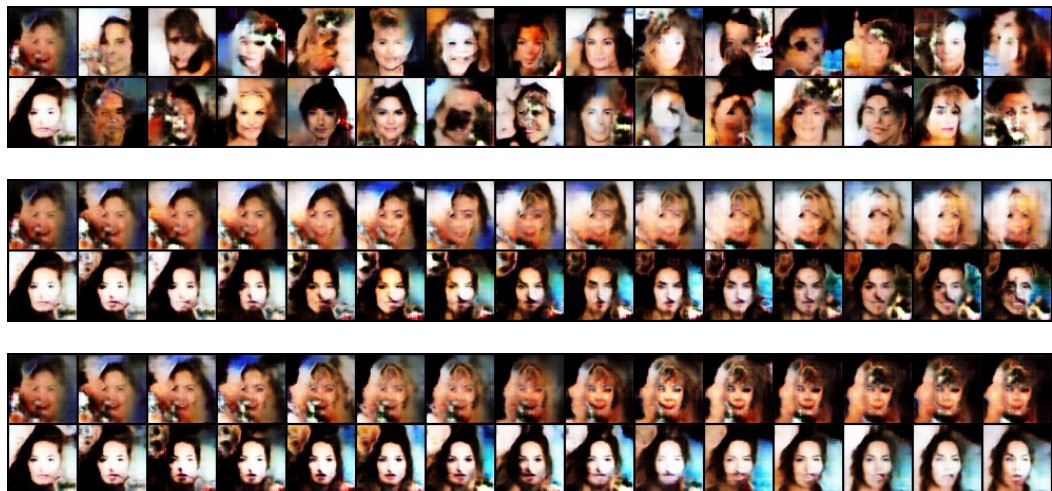

Figure 6: Visualization of the Markov chains of MH-GAN (top), DDLS (middle), and REP-GAN (bottom) on CelebA with WGAN backbone.

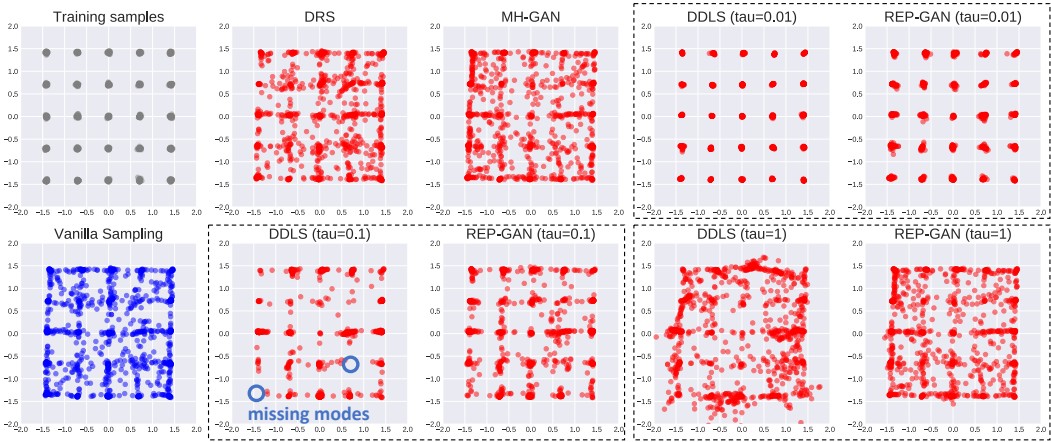

Figure 7: Visualization of samples with different sampling methods on the 25-Gaussians dataset. Here $\tau$ denotes the Langevin step size in Eqn. (17).

that some modes are missed. When $\tau = 1$, samples of DDLS diverge far beyond the 5x5 grid. In comparison, our REP-GAN does not suffer from these issues with the MH correction steps accounting for the bias introduced by numerical errors.

**Scale to more modes.** In the above, we have experimented w.r.t. a relatively easy scenario where the multi-modal distribution only has 5x5 modes ($n = 5$ modes along each axis). In fact, the distinctions between the sampling methods become even more obvious when we scale to more modes. Specifically, as shown in Figure 8, we also compare them w.r.t. mixture of Gaussians with 9x9 and 13x13 modes, respectively. The rest of the setup is similar to 25-Gaussians. Note that throughout the experiments in this part, we adopt proper step size, $\tau = 0.01$, for the gradient-based methods (DDLS and REP-GAN) by default.

Under the more challenging scenarios, we can see that the gradient-based methods still consistently outperforms MH-GAN. Moreover, our REP-GAN has a more clear advantage over DDLS. Specifically, for 9x9 modes, our REP-GAN produces samples that are less noisy (i.e., less examples distinct from the modes), while preserving all the modes. For 9x9 modes, DDLS makes a critical mistake that it drops one of the modes (left down corner, marked with red circle) during the Markov chain update. As discussed above, we believe this is because DDLS has a bias towards regions with high

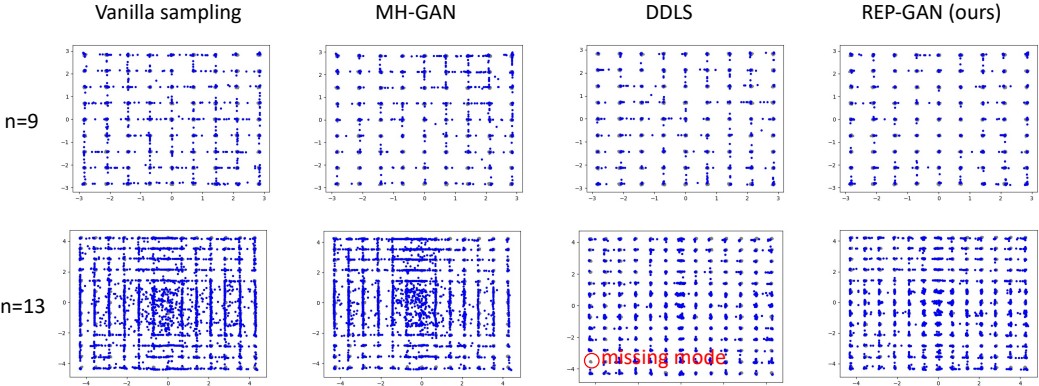

Figure 8: Visualization of the multi-modal experiments with more modes. Specifically two cases are considered, 9x9, and 13x13 mixture of Gaussian. The true data points are shown in grey in background, and the generated points with different sampling methods are shown in blue.

probability, while ignoring the diversity of the distribution. In comparison, REP-GAN effectively prevents such bias by the MH correction steps.

