# OpenReview forum: "Efficient Sampling for Generative Adversarial Networks with Reparameterized Markov Chains"
_ICLR.cc/2021/Conference — Reject_

### Official Review · AnonReviewer4 · 2020-10-27
**Neat paper!**

**Rating:** 7
**Confidence:** 4

**Review:**

**Summary**
The paper proposes an MCMC based sampling mechanism for GANs. In contrast to earlier work, the proposal distribution is conditioning conditioned on the previous state (here in latent space), which is supposed to help sampling efficiency. This is achieved by a clever re-parametrization of intermediate steps of the MCMC chain. As an example, the authors provide a Langevin version (which uses gradient information) of their method.

I enjoyed reading this well-written paper and think the re-parameterized MCMC chain is a very neat idea that fits very naturally in the GAN framework. The paper is of quite incremental nature though: little to no theory contribution, only some incremental empirical benefits. I also have some doubts whether the comparisons are fair in terms of compute and effective sample size (more below).

Below are a few points to consider:

**Effective sample size**
What puzzles me here is the fact that while the independent proposal of Turner et al gets stuck in certain parts of the space, the Langevin updates of the proposed method (Figure 4) do have less variability. I assume the incremental improvement happen due to relatively small step sizes in the Langevin sampler? This would have a bad effect on the effective sample size. Does the MCMC chain eventually visits other parts of the space, or does it need to be restarted for every sample created? This does not seem to be true for the MH sampler, which despite its low acceptance rate produces samples from the entire space. Could the authors clarify this, and if possible provide measure of effective sample size for a fixed number of iterations?

**Compute**
The authors do neither discuss nor evaluate the computational load compared to other methods, especially performance metrics as a function of compute would be interesting. Or maybe the computational load is comparable for all methods?

**Title**
The term "coupled" is currently heavily used within the MCMC community, e.g. [2], and it denotes a different technique. I would suggest replacing it by "re-parameterized" which would be more accurate in that respect.

**"Proof" for convergence**
The abstract claims that convergence to the true data distribution is proved, but that is really stretching the term. All the paper contains is a valid MH ratio. But that alone does not prove convergence, for which the authors would need to go into the weeds of MCMC convergence, with conditions on the target distribution. This is especially true for the Langevin proposal, e.g. [1].

**Experiments**
I would suggest to move the Swiss roll dataset to the Appendix and replace it with something more interesting, preferably higher dimensional. This would make the paper stronger. In fact some analysis how all the different sampling methods compare as a function of dimensional (or "difficulty") would make it even more so.


[1] Roberts, G. O., & Tweedie, R. L. (1996). Exponential convergence of Langevin distributions and their discrete approximations. Bernoulli.

[2] Jacob, P. E., O'Leary, J., & Atchadé, Y. F. (2017). Unbiased markov chain monte carlo with couplings. JRSS-B.

---

> ### Author Response · Authors · 2020-11-20
> **Response to Reviewer #4**
>
> Thank you for your valuable comments. We address them in detail as follows.
>
> ---
> **Q1**: Improvements are due to relatively small step sizes in the Langevin sampler? The effect on the effective sample size? Does the MCMC chain eventually visits other parts of the space？
>
> **A1**: First of all, the improvement is brought by the relatively small step size that we choose for better sample quality. As is mentioned in Section 5.2, we evaluate our methods following the conventional setting of the sampling methods of GANs (as in MH-GAN, DDLS) by running a chain for 640 steps, and taking only the sample of the last step for evaluation. In this test scenario, because we only need the last example, we do not need the chain to have diversity among its different steps, thus effective sample size is not a matter of concern here. Instead, we mainly use the chain to finetune the initial sample with a relatively small step size (e.g., tau=0.01 for CIFAR-10) for better sample quality.
>
> As shown in the newly added Figure 5, our gradient-based proposal can also propose very diverse examples with large step size (tau=1 for CIFAR-10), just like the independent proposal of MH-GAN (which is an extreme case of our REP proposal). Therefore, the resulting chains are likely able to visit other parts of the space.
>
> We have added this discussion to the last paragraph of Section 5.2. We have also added multi-modal experiments in Appendix A.4, and show that our method well preserves the sample diversity while promoting the sample quality.
>
>
> ---
> **Q2**: Comparison of computation cost.
>
> **A2**: We have added a comparison of computation cost in Table 5 in the Appendix A.3. Comparing DDLS and ours, they take 88.94s and 88.85s to run 500 chains for 640 steps on a single 1080 Ti GPU, respectively, hence the difference is negligible. Without the MH-step, our method takes 87.62s, meaning the additional MH-step only costs 1.4% computation overhead, which is almost negligible, but it brings a significant improvement in sample quality.
>
>
> ---
> **Q3**: The title needs to be changed to avoid confusion with existing literature.
>
> **A3**: Thank you for your kind advice. We are happy to take your excellent advice and rename the title with "Reparameterized Markov Chains" together with the short name "REP-GAN".
>
>
> ---
> **Q4**: The abstract claims that convergence to the true data distribution is proved, but that is really stretching the term. All the paper contains is a valid M2H ratio. But that alone does not prove convergence.
>
> **A4**: Thanks for pointing it out. Indeed, as our reparameterized proposal is designed to be a general framework compatible with many MCMC proposals, we cannot give convergence results without knowing the properties of the specific proposal distribution. Hence, in our Theorem 1, we assume the chain having certain regularity conditions (irreducible, aperiodic, and not transient [1]) to obtain convergence in general. As you suggested, we have revised our statements in the paper to make it more clear and rigorous.
>
> [1] Gelman et al. (2013). Bayesian data analysis. CRC press.
>
>
> ---
> **Q5**: Compare different methods as a function of input dimension, or task difficulty.
>
> **A5**: Following your suggestions, we compare the sampling methods with increasing task difficulties. Specifically, in the multi-modal experiments in Appendix A.4, we experiment with the mixture of Gaussian distributions that have 5x5, 9x9, and 13x13 modes. Intuitively, the tasks get harder with more modes to cover, as it becomes more challenging for the sampling methods to improve sample quality while preserving sample diversity. As shown in Figure 7 and 8, our method has done a good job by outperforming both independent samplers (MH-GAN) and pure Langevin steps (DDLS). It shows our method is less unlikely to degrade sample diversity while promoting the overall sample quality. Please refer to the discussion there for more details.
>
>
> ---
> Overall, thank you a lot for your insightful comments, which encourage us to collect more interesting results. Hope our feedback could help answer your concerns!

---

### Official Review · AnonReviewer2 · 2020-10-29
**good idea but not convincing experiments**

**Rating:** 5
**Confidence:** 4

**Review:**

Summary:

The paper proposes an MCMC sampling strategy for GANs. The idea is clear: for high-dimensional x, making a good proposal is difficult, so they propose to do that in the latent space.  Then they use a similar strategy as MH-GAN to compute a rejection strategy. The difference between the two methods is that proposal in MH-GAN does not depend on x while the proposed method does, and the argument is that it results in a higher acceptance rate.

The idea is interesting, and the paper is relatively well-written, but the execution and experiment left a lot to be desired.

* One of the arguments of the paper (in the related work) the method overcomes sample low sample diversity. No metric is reported for that. The authors need to report LPIPS [1] and compare different methods.

*  Fig 4 shows very poor diversity; all samples seem identical to each other, and honestly, it seems that the GAN collapsed.

* The results in Fig 3 is also very strange. First of all, what is the point of Fig 3 (left)? All IS scores are roughly going up, but this figure is no comparison and does not show any if one method is doing better.  Fig 3 (right) is more concerning. One expects that as the epoch increases, the generator becomes better at generating samples. The acceptance rate improves (b/c G generates more realistic results and discriminator cannot tell them apart); why the acceptance rates are going down?

* Fig 2 is also presented in a confusing way. All methods except the second column from left almost perform the same and propose methods is in both columns (!!) What does each column, row mean? Given what I read in the related work, I expected to see the model's performance in the sample a multi-modal distribution. In fact, I want to see that: 100 Gaussian distributions centered at the 10x10 grids and small enough variance. I want to see how this method performs in recovering centers of the clusters (you can see examples in the literature).

* Given that finally, the sampling strategy becomes very similar to the recent energy-based method. It is fair to say comparison is required. Yes, GAN is and the energy-based methods are different but the sampling strategy (the Langevian dynamic) are very similar and I would like to see how those two methods perform in term of quality of the samples.


* Finally, if one views this method as a general strategy for generating samples using GAN, perhaps you can think of this approach as an approximate inference approach for inference in a complicated Baysian model. This paper does not focus on that and I don't expect authors to do any experiment for that but it can useful there.



[1] Richard Zhang, Phillip Isola, Alexei A Efros, Eli Shechtman, and Oliver Wang. The Unreasonable
Effectiveness of Deep Features as a Perceptual Metric. In CVPR, 2018.

---

> ### Author Response · Authors · 2020-11-20
> **Response to Reviewer #2 (2/2)**
>
> **Q4**: Figure 2 is presented in a confusing way. Expect a multi-modal experiment.
>
> **A4**: Sorry for the confusion. Each sub-figure shows the samples obtained by the corresponding sampling method on its top. The sub-figures are arranged left to right, and we show pairs of results of DDLS and our method under different step sizes (tau) to illustrate their different behaviors, now marked with dashed boxes.
>
> To address your concern, we have added the multi-modal experiments, including a  25-Gaussians experiment as well as more difficult tasks with more modes, in Appendix A.4. In the newly added Figure 7 and 8, our method shows clear benefits in terms of sampling quality, sample diversity, and numerical stability. Please refer to the discussion there for more details.
>
>
> ---
> **Q5**: Similarity and comparison with energy-based models (EBM).
>
> **A5**: Yes our sampling scheme of GANs is similar to that of EBM because essentially they both utilize HMC-like methods. However, there is an essential difference that EBM uses HMC sampling for training, while our method only uses it to post-process trained GANs. Thus, the difference between their performance could be largely due to the difference of training mechanisms between EBM and GAN themselves, instead of the difference in sampling methods. As far as we can see, EBM and GAN have become two strong competitors in the field of generative models, each with a lot of modern variants with state-of-the-art results. A fair comparison of the two diagrams may need a more thorough, large-scale study of various components that affect their relative performance, which is left for our further work.
>
>
> ---
> **Q6**: Our method can serve as a general approximate inference method for complex Bayesian models.
>
> **A6**: Yes, it is! Thanks for pointing out that. Indeed, our method can serve as a general approximate inference technique for Bayesian models by bridging MCMC and GANs. Previous works [2,3,4] also propose to avoid the bad geometry of a complex probability measure by reparameterizing the Markov transitions into a simpler measure. However, these methods are limited to explicit invertible mappings without dimensionality reduction.
> In our work, we first show that it is also tractable to conduct such model-based reparameterization with implicit models like GANs.  Also notice that our method is designed for GANs, but not limited to existing (unstable) GAN training mechanisms because our method does not depend on how GANs are trained. In fact, our method can be applied to general implicit generative models with density ratio estimation.
>
> We have now added this discussion to the last paragraph in Section 4.2.
>
> [2] Marzouk et al. (2016). An introduction to sampling via measure transport.
>
> [3] Hoffman et al. (2019). Neutra-lizing bad geometry in Hamiltonian Monte Carlo using neural transport.
>
> [4] Titsias. (2017). Learning model reparametrizations: Implicit variational inference by fitting MCMC distributions.
>
>
> ---
> Thank you again for all your insightful comments and hope our feedback is helpful!

---

> ### Author Response · Authors · 2020-11-20
> **Response to Reviewer #2 (1/2)**
>
> Thanks for your valuable feedback. We address them in details as follows.
>
> ---
> **Q1**: The paper claims in the related work that it overcomes low sample diversity, but no metric is reported for that.
>
> **A1**: Sorry for the confusion. In the related work, we actually do not claim that our method aims for (or has solved) the low sample diversity problem (i.e., mode collapse problem), but just point out that mode collapse is one of the failures of GANs, and lots of papers work on it. Instead, we claim that our method belongs to another thread of works that use sampling methods to improve overall sample quality, and our methods improve sample efficiency over previous ones. We have edited the discussion in the related work to avoid confusion.
>
> As for LPIPS score, as discussed in [1], it is not very suitable for the dataset we study here, CIFAR-10 and CELEBA, because the image resolution is very low. But, we still followed your suggestions and calculated the LPIPS scores with 10k samples by the official code and reported here. Note that these scores are too close compared to that on other common datasets used for evaluating LPIPS. Besides, there are some strange phenomena, for example, DDLS demonstrates the best sample diversity here. It is even higher than the real data on CELEBA. While in fact, it performs worse than others in terms of Inception scores. Hence LPIPS here may not correlate well with the overall sample quality of CIFAR-10 and CELEBA.
>
>
> |CIFAR10 | LPIPS|
> |--------|--------|
> |Real data | 0.49379 +/- 0.00076|
> |GAN | 0.46634 +/- 0.00074 |
> |MHGAN | 0.46508 +/- 0.00075|
> |DDLS | 0.49098 +/- 0.00100|
> |Ours | 0.46657 +/- 0.00076|
>
> |CELEBA | LPIPS|
> |--------|--------|
> |Real data | 0.28959 +/- 0.00074 |
> |GAN | 0.28835 +/- 0.00073 |
> |MHGAN | 0.28807 +/- 0.00072 |
> |DDLS | 0.42161 +/- 0.00091 |
> |Ours | 0.28980 +/- 0.00074 |
>
> Apart from this result, we have added synthetic multi-modal experiment results in Appendix A.4, and show our method preserves sample diversity while greatly promoting sample quality.
>
> [1] Mao et al. (2019). Mode seeking generative adversarial networks for diverse image synthesis. CVPR.
>
> ---
> **Q2**: Fig 4 shows very poor diversity, and the chain collapses.
>
> **A2**:  As shown in the newly added Figure 5, our gradient-based proposal can also propose very diverse examples with large step size (tau=1 for CIFAR-10), just like the independent sampler of MH-GAN, and the chain won’t appear to be stuck. As is mentioned in Sec. 5.2, we evaluate our methods following the conventional setting of the sampling methods of GANs (as in MH-GAN, DDLS) by running a chain for 640 steps, and taking only the sample of the last step for evaluation. In this test scenario, because we only need the last example, we do not need the chain to have diversity among its different steps. Instead, we mainly use the chain to finetune the initial sample with a relatively small step size (e.g., tau=0.01 for CIFAR-10) for better sample quality. We have also included this discussion to the last paragraph in Section 5.2.
>
> ---
> **Q3**: What is the point of Figure 3 (left)? Why are the acceptance rates going down in Figure 3 (right)?
>
> **A3**: From our view, we think Figure 3 (left) is an important result demonstrating the effectiveness of our method. As shown in Figure 3 (left), the training process of GANs is very unstable, and the inception scores vary a lot from epoch to epoch. To comprehensively evaluate our method, we show the training process of GANs. As shown in Figure 3 (left), our method outperforms previous methods consistently (superior in most epochs) and significantly (improvement > error bar).
>
> As for the confusing behavior in Figure 3 (right), we also noticed it and thought it is an interesting phenomenon worth exploring. A possible explanation is that, as the training continues, the distribution landscape becomes more and more complex to match the high-dimensional data distribution. As a result, with constant sampling step size as previous epochs, the proposed samples may become more and more dissimilar to the current one, which may result in a lower acceptance ratio.
>
> We have added these discussions to the second paragraph in Section 5.2.
>
> (See more below)

---

> > ### Comment · AnonReviewer2 · 2020-11-24
> > **the goal of the paper needs to be clear**
> >
> > I am not really sure I understand this statement:
> >
> > "...our method belongs to another thread of works that use sampling methods to improve overall sample quality, and our methods improve sample efficiency over previous ones..."
> >
> > The acceptance rate is one criterion for sample quality. Consider a chain that only samples from one of the picks of multimodal distribution. The acceptance rate is probably very high but the chain doesn't cover the whole distribution. This is why a metric for diversity is needed. I understand that LPIPS doesn't show a significant variation between different methods in your experiment and I am open to another suggestion. We need to have a metric to show that it produces better sample quality. The other toys exa
> >
> > The method is nice but the experiment does not show where the advantage is (beyond the toy example). This is also pointed out by one of the reviewers.

---

> > > ### Author Response · Authors · 2020-11-25
> > > **Inception scores have taken sample diversity into consideration**
> > >
> > > Thanks for your reply. By writing "...our method belongs to another thread of works that use sampling methods to improve overall sample quality, and our methods improve sample efficiency over previous ones...", we are suggesting that our method focus on the overall sample quality, e.g. measured by Inception scores, like previous sampling methods for GANs, instead of focusing specifically on the diversity problem, e.g. measured by the number of covered modes, like previous works fixing mode collapse. Compared to previous sampling methods, our work improves both sample quality (measured by Inception scores) and sample efficiency (measured by average acceptance ratio) by adopting a dependent reparameterized proposal.
> > >
> > > As you have pointed out, a high acceptance ratio does not necessarily lead to high sample quality, we also need to take sample diversity into consideration. But as mentioned in our response to your Q2, the conventional evaluation protocol of the sampling methods of GANs only require the last example of the chain, thus, **we do not need diversity among different steps of one chain, but only require sample diversity of the final samples collected from many different chains.**
> > >
> > > To measure the overall sample quality of these samples, we adopt Inception score, a widely adopted metric for GANs, following previous sampling methods of GANs. We note that **the Inception score does take sample diversity into consideration.** Technically, Inception scores calculate the exponential mutual information between X and Y, i.e. exp(I(X, Y)). The mutual information admits the decomposition I(X,Y) = H(Y) - H(Y|X), that is to say, **a large Inception score promotes the overall entropy of labels across the sample set (high diversity), and decreases the uncertainty of labels given a specific sample (better quality per-sample).**
> > >
> > > Therefore, we believe that the Inception score is a good metric for measuring the overall sample quality (diversity between samples + quality per sample).
> > >
> > > ---
> > > Thank you again for your reply and please let us know if you have additional questions.

---

> ### Author Response · Authors · 2020-11-24
> **Any additional questions?**
>
> Dear AnonReviewer 2,
>
> Thanks again for reviewing our paper. Please let us know if you have any additional questions or require further clarifications. We are happy to address them before the rebuttal ends.

---

### Official Review · AnonReviewer1 · 2020-10-29
**A great solution but I'm not yet fully convinced by the problem it is solving**

**Rating:** 5
**Confidence:** 4

**Review:**

### Summary
In this work, the authors propose coupling markov chain GAN (CMC-GAN), a technique that allows to draw GAN samples by running an MCMC chain that uses the generator to produce proposals, and the discriminator to compute acceptance probabilities.

Whereas prior work by Turner et al. (2019) uses an independent proposal, for each sample, the present work uses a small random perturbation of the last sample *in the latent space* to create the proposal.
Goodfellow et al. 2014 showed that in the large data limit and large model capacity limit for absolutely continuous distributions, the output of the optimal discriminator represents encodes the density ratio of target and generated data.
Using this result, the authors of the present work derive a tractable expression for the acceptance probabilities that leads to an unbiased estimate of the target distribution in settings where the results of Goodfellow et al. 2014 hold.
They also show that an analogue acceptance criterion can be derived for the popular WGAN.
The authors show experimental results on the synthetic swiss roll dataset as well as CIFAR10 and CELEBA with DCGAN and WGAN architecture, showing improved sample quality (measured by IS on the image datasets) compared to standard GAN and competing MCMC approaches, and improved acceptance rate compared to GAN.

### Decision
I believe that proposes the most natural way for performing MCMC sampling on GANs. Both the dependent sampling based on the latent space and the acceptance mechanism based on the discriminator are natural and clean and, I'm sure will end up useful one way or another.
Where the paper is lacking, in my opinion, is in making a case *why* and *when* one should be using MCMC based methods in the first place when using GANs.
While I do find the results on the swiss roll experiments are convincing, I am not convinced by the results on image datasets.
The improvements seem to be minor, inconsistent, and on baselines performing worse than commonly reported in the literature.
Given how clean the proposed method itself is authors should improve the motivation by trying to find a more convincing application. I think it would be a wasted opportunity for what could be a very nice paper to accept it without such improvements.

### Questions/Suggestions

- The only motivation for the use of MCMC techniques is to not "waste the discriminator". Why should I expect the discriminator to be any better than the generator that it was trained with? Are there situations where one would naturally end up with a "better" discriminator than generator? Given that ordinary GAN sampling has 100 percent sample efficiency, I feel that a stronger argument in favor of using any MCMC based-technique in the first place is necessary.

- While I do find the experiments on swiss roll convincing, I am confused by the experiments on image GANs. **First**, the inception scores given in Table 2 seem significantly below those usually reported in the literature. **Second**, the experiments in Figure 3 show a minimal improvement of CMC-GAN compared to ordinary GAN sampling, in particular compared to the wide oscillation of IS over the different epochs. In particular, Figure 3 and Table 2 do not seem to be consistent. **Third**, it is my understanding that it does not make much sense to evaluate the IS on CELEBA. Usually, FID is used instead.

- Recent work such as [Berard et al.](https://arxiv.org/abs/1906.04848), [Schafer et al.](https://arxiv.org/abs/1910.05852),  [Arjovsky and Bottou 2017](https://arxiv.org/abs/1701.04862) suggests that the behavior of discriminator training on image GANs might be radically different than in the idealized case analyzed by Goodfellow et al. in 2017.
Could this be the reason why the results on image data sets fall short of those obtained in the absolutely continuous swiss roll dataset. Maybe techniques such as in [Dieng et al. 2019](https://arxiv.org/abs/1910.04302) could be useful?

- Although it has a somewhat different goal, it might still be useful to cite [Song et al.](https://arxiv.org/abs/1706.07561).

=============================================================================================

After discussion with the authors and reading the other reviews I still believe that the paper should not be accepted and therefore stay with my original review. While the authors have shown improvement over previous work (MH-GAN) using a very natural idea, this previous work in turn has not provided sufficient evidence that MCMC-GANs are useful.
I believe that we should not accept further MCMC-GAN papers, before this methodology has shown any improvement on a plausible use-case.

---

> ### Author Response · Authors · 2020-11-20
> **Response to Reviewer #1 （2/2)**
>
> **Q4**: Figure 3 and Table 2 seem to be inconsistent.
>
> **A4**: Sorry for the confusion. First, Figure 3 only demonstrates the first 25 epochs of training, while in total, we train the model for 60 epochs. Second, as mentioned in the caption of Table 2, the results of GAN, DRS, MH-GAN are directly taken from the MH-GAN paper (Turner et al., 2019), while the results in Figure 3 are based on our re-implementation to compare results epoch by epoch. So they are different because of the instability of GANs. To see the comparison of these methods with the same implementation, we recommend Table 3, where scores are all reported based on our implementation, and you can find the corresponding versions of MH-GAN (IND proposal w/ MH) and DDLS (REP proposal w/o MH) there. Our method still outperforms them in most cases.
>
> ---
> **Q5**: It does not make much sense to evaluate IS on CELEBA. Should use FID instead.
>
> **A5**: We use Inception scores for evaluation because previous works usually report Inception scores on these two benchmark datasets, CIFAR-10 and CELEBA. For a fair comparison, we also report Inception scores following the same setup.
>
> Nevertheless, we think your concern is reasonable as CELEBA is quite different from ImageNet and it is better to utilize the real data statistics here to measure sample quality. Thus, we additionally report the comparison of FID scores in Table 4 (Appendix A.3) based on our re-implementation (because there is not any FID reported on these benchmarks in previous sampling methods) on all benchmarks. It can be seen that our method also outperforms previous ones in terms of FID scores.
>
> ---
> **Q6**: The behavior of discriminator training can be radically different than in the idealized case, and this may explain the limited improvement on image data. What mechanism might help?
>
> **A6**: We agree with you that this could be the reason why the sample quality of sampling methods is still lower than that of real data distribution, and even longer chains cannot fully mitigate the gap. In the sampling methods of GANs, we always assume the discriminator gives a perfect estimation of the density ratio for the theory to hold. However, it is usually not the case in practice because the generator is also changing all the time, and the one-step update of the discriminator cannot fully capture this information, but it can capture a certain extent information of density ratio which explains why the sampling methods can consistently improve over the baseline at each epoch.
>
> From our view, the estimated density ratio is enough to push the generator better but not able to bring it up to the data distribution. Hence, how to develop mechanisms that bring more accurate density ratio estimation remains an interesting research direction. We are willing to put more effort into investigating this problem in the future.
>
> We have added this discussion in the second assumption in Appendix A.1.
>
> ---
> **Q7**: Helpful to cite the paper “A-NICE-MC”.
>
> **A7**: Thanks for your advice. A-NICE-MC is an excellent pioneer work on bridging MCMC and NNs by learning NN transition kernels. Ours instead directly utilizes properties of existing NN modules for the transition kernel. The two share the same spirit that NN helps design MCMC transition kernels. We have added the citation in the related work.
>
>
> ---
> In all, thanks for your valuable review, and we hope you find our explanations helpful!

---

> > ### Comment · AnonReviewer1 · 2020-11-23
> > **I am still not convinced of the benefits of combining MCMC with GANs**
> >
> > Thank you for the replies. My main concern is still that the paper does not convince me that combining MCMC with GANs is helpful in high-dimensional settings (which form the main applications of GANs in the first place).
> >
> > The fact that the authors method of combining MCMC with GANs  improves over previous such attempts (MH-GAN) does not address this concern.
> > Even if the GAN baseline were to achieve a reasonable inception score, the improvements are too minor and inconsistent (accross epochs) to warrant, in my opinion, the overhead of adding an MCMC sampling step, since sampling from a GAN is trivial, as-is.
> > On that note, I would also like to ask what randomness  the confidence intervals are referring too. Given the large variance accross iterations it seems like they are not referring to multiple runs?
> >
> > The above concerns are aggravated by the fact that the performance of the GAN baseline is worse than that of common basic implementations. I agree with the authors that they outperform MH-GAN significantly and as I wrote in my review I believe that the authors get MCMC-GANs "right". However, neither the authors work nor the prior work on the subject convinces me that MCMC-GANs are worth considering in the first place.
> >
> > As long as an at least somewhat convincing use-case is lacking, I advise against accepting this work to top-tier ML conferences.

---

> > > ### Author Response · Authors · 2020-11-24
> > > **Further Response to Reviewer #1 (2/2)**
> > >
> > > As for your concern on the improvements (minor and inconsistent across epochs), we make more detailed explanations here:
> > >
> > > ---
> > > + The shaded error bar in Figure 3 (left) reports the deviation of Inception scores. The [Inception scores](https://github.com/openai/improved-gan) calculate the mean and standard deviation of Inception scores with 10 splits of the total samples. The standard deviation stands for the variation of Inception scores due to sampling.
> > > + As we know, the training process of GANs is very unstable, and its performance can drop down and rise up quickly across epochs. Therefore, the quality of the learned discriminator can also vary a lot from epoch to epoch, which results in the outliers in Figure 3 (left). But this is also why we report Inception scores per epoch in Figure 3 (left), because we show that our method can deliver consistent improvements as long as we have a moderately good discriminator.
> > > + We still believe that the improvements shown in Figure 3 (left) are significant and consistent across epochs. To begin with, there are only 5 epochs in the total 25 epochs where the sampling methods perform worse than GANs, and 4 of the 5 epochs are before the 10th epoch when the training is very insufficient. After the 15th epoch, the sampling methods always outperform the vanilla sampling of GANs significantly, often with a large improvement of Inception scores, e.g. more than 0.4 (>10%) in the 19th epoch. Thus we believe it is still an effective approach as it can improve the sample quality easily and significantly in most cases without the need to re-train the models.
> > > + The synthetic experiments shown in Figure 2, 7, 8, including the Swill roll and 25-Gaussians, show more clear advantage of the sampling methods. For example, in 25-Gaussians, because the generators of GANs are typically continuous functions, they lend to produce more samples in the overlapping regions between modes, while those examples are of very low probabilities in the data distribution. Figure 2 and 7 show the sampling methods can successfully alleviate this kind of artifacts by utilizing the density ratio information in the discriminator.
> > > ---
> > >
> > > Thank you again for your feedback and hope our explanations are helpful!
> > >
> > > [1] Abbasnejad et al. (2019). A Generative Adversarial Density Estimator. CVPR.
> > >
> > > [2] Sen et al. (2018). Mimic and classify: A meta-algorithm for conditional independence testing.
> > >
> > > [3] Bellot et al. (2019). Conditional Independence Testing using Generative Adversarial Networks. NeurIPS.
> > >
> > > [4] Kliger et al. (2018). Novelty Detection with GAN.
> > >
> > > [5] Li et al. (2018). Anomaly Detection with Generative Adversarial Networks for Multivariate Time Series.
> > >
> > > [6] Yu et al. (2017). SeqGAN: Sequence Generative Adversarial Nets with Policy Gradient. AAAI.
> > >
> > > [7] Lunz et al. (2020). Inverse Graphics GAN: Learning to Generate 3D Shapes from Unstructured 2D Data.
> > >
> > > [8] Li et al. (2018). Point Cloud GAN. ICLR workshop.
> > >
> > > [9] Song et al. (2017). A-NICE-MC: Adversarial Training for MCMC. NeurIPS.
> > >
> > > [10] Patel and Oberai. (2019). Bayesian Inference with Generative Adversarial Network Priors. NeurIPS workshop.
> > >
> > > [11] Marzouk et al. (2016). Sampling via Measure Transport: An Introduction. Handbook of Uncertainty Quantification. pp. 1-41.
> > >
> > > [12] Hoffman et al. (2019). Neutra-lizing Bad Geometry in Hamiltonian Monte Carlo Using Neural Transport.
> > >
> > > [13] Titsias. (2017). Learning Model Reparametrizations: Implicit Variational Inference by Fitting MCMC Distributions.

---

> > > > ### Comment · AnonReviewer1 · 2020-11-24
> > > > **Variance of results**
> > > >
> > > > ### CIFAR10 results:
> > > >
> > > > It is true that GAN training tends to be unstable. However, comparing the Plots in figure 3 to my own recollections of training standard GAN variants like DCGANs, the experiments of the present paper still seem to be very unstable.
> > > >
> > > > Given the instability of training, the more meaningful "standard deviation" would be the one obtained by independent reruns of the experiments with different random seeds. Based on the per-epoch variance shown in figure 3, this standard deviation is expected to be *much* larger than the standard deviation of computing the inception score.
> > > > The improvements over regular GAN due seem to be well within the standard deviation of the results between epoch 20 and 25, which is why I strongly disagree with the author's claim that they established a significant improvement of MCMC sampling over regular GAN.
> > > >
> > > > ### Other applications of GANs:
> > > >
> > > > I agree with the authors that there are other applications of GANs than images and it seems very plausible that there exist meaningful use cases for MCMC-GAN fusion. I highly encourage the authors to resubmit their work after adding a case study on such an application.
> > > > As it is, the work only establishes improvement over standard GAN low-dimensional toy examples. I stand by my opinion that at this state of affairs the paper should not be accepted to ICLR.

---

> > > > > ### Author Response · Authors · 2020-11-25
> > > > > **About error bar calculation**
> > > > >
> > > > > Thanks for your reply. We address your concerns as follows:
> > > > >
> > > > > ---
> > > > > Q1: CIFAR-10 results seem very unstable. The reported error bar should be averaged over different runs instead.
> > > > >
> > > > > A1: In our first reply to your Q2, we have mentioned that our implementation directly adopts the [official code of MH-GAN](https://github.com/uber-research/metropolis-hastings-gans).  As shown in Figure 5(a) of the [MH-GAN paper](https://arxiv.org/pdf/1811.11357.pdf), **their reported GAN training is also very unstable**. Meanwhile, **in their figure, the shaded error bar is also calculated w.r.t. the variance of Inception scores across different splits of samples, instead of averaged over multiple runs of GANs**. We agree with you that it is helpful to report results of different runs, but since the GAN training is so unstable here, simply averaging over multiple runs might erase anything meaningful. It can be seen that our Figure 3 (left) is an analogy of Figure 5(a) of MH-GAN. The two figures demonstrate similar behaviors, while our method enjoys superior Inception scores to MH-GAN in most cases.
> > > > >
> > > > > ---
> > > > > Q2: Applying to other applications of GANs.
> > > > >
> > > > > A2: Thanks for your suggestions. In our last reply, we list various applications of GANs to highlight that our method is general and potentially helpful to obtain a more accurate estimation for various scenarios. Nevertheless, in the first place, our work belongs to the sampling methods of GANs and we have shown superior empirical results on the common benchmarks of sampling methods. We also believe it would be more convincing to apply our method to general problems beyond images. However, we were not able to finish the experiment on new tasks in the limited time. We are definitely happy to add more results to our final version.

---

> > > ### Author Response · Authors · 2020-11-24
> > > **Further Response to Reviewer #1 (1/2)**
> > >
> > > Thanks for your reply and positive feedback for our additional discussions on the experiments. As for your left concern about the benefits of combining MCMC with GANs, we would like to highlight some of our contributions:
> > >
> > > ---
> > > + Although GANs are developed for generating images in the first place, they now find successful applications in many other scenarios. For example, GANs can serve as a novel, effective, scalable method for many statistical problems for high-dimensional data, including Bayesian inference (explained below), density estimation [1], hypothesis testing [2,3], outlier detection [4,5], etc. GANs are also applied to various kinds of data beyond images, e.g. time series [5], text [6], mesh grid [7], point cloud [8], etc. Our method is general and applicable for these variants of GANs to produce more accurate estimation with the combination of MCMC sampling, as the Reviewer #2 pointed out that our method can serve as the “approximate inference approach for inference in a complicated Baysian model”.
> > > + There are previous works combining MCMC and GANs as discussed in Related Work and Section 4.2. A-NICE-MC [9] trains a neural proposal for MCMC with GANs, while [7] adopts GANs for the prior distribution of Bayesian inference with MCMC. With regard to our proposed method, there are some previous works [11,12,13] that avoid the bad geometry of a complex probability measure by reparameterizing the Markov transitions into a simpler measure. However, these methods are limited to explicit invertible mappings without dimensionality reduction. In our work, we first show that it is also tractable to conduct model-based reparameterization for implicit models like GANs. In fact, our method can be applied to general implicit generative models with density ratio estimation, and offer more accurate estimation for the statistical problems to solve.
> > > + Theoretically, it is impossible for the continuous generator of GANs to model distributions with discontinuous support regions, while our MCMC methods can overcome this limitation with the acceptance-rejection (MH) steps. Thus our sampling methods make it possible to close the gap between the generative distribution and the discontinuous data distribution.
> > >
> > > (See more below)

---

> ### Author Response · Authors · 2020-11-20
> **Response to Reviewer #1 （1/2)**
>
> Thanks for your valuable comments. We address them in detail as follows.
>
>
> ---
> **Q1**: Why should I expect the discriminator can be any better than the generator that it was trained with? When and why one expects to use MCMC methods for GANs?
>
> **A1**: In the original minimax formulation of GANs, given a discriminator D in the outer loop, the generator G should optimize the inner loop exactly, but in practice, G typically takes only one gradient step, thus, it has not fully cultivated the density ratio information in the dynamically changing discriminator. Therefore, given a trained GAN, it is still possible to boost the performance of the generator with iterative fine-grained methods (such as MCMC sampling) that fully utilize the density ratio information contained in the discriminator.
>
> Therefore, we can see sampling methods are suitable for scenarios when we wish to further improve the generation quality of a trained generator. As shown in our experiments, it works for different GANs and datasets, and is orthogonal to other kinds of methods to improve GANs, such as new architecture modules and training techniques. The method is also plug-and-play because it does not involve dependence on any specific generator module, but only depends on the latent input and the discriminator output. Hence we can apply it to many variants of GANs to improve their performance by merely regarding them as blackboxes.
>
> We have added these discussions in the second paragraph of Introduction.
>
>
>
> ---
> **Q2**: Inception scores reported are lower than literature.
>
> **A2**: Sorry for the confusion. In fact, as mentioned in Table 2, the reported results of GAN, DRS, and MH-GAN are taken directly from the MH-GAN paper (Turner et al., 2019). Because DDLS does not report results on these common benchmarks, we report our re-implementation results instead.
>
> Our implementation directly uses [MH-GAN’s official code](https://github.com/uber-research/metropolis-hastings-gans) with no modification on their hyper-parameters. And MH-GAN’s implementation of its backbone, DCGAN, directly follows [PyTorch’s official example of DCGAN](https://github.com/pytorch/examples/tree/master/dcgan) without modification on hyper-parameters. The hyperparameters here are different from the original DCGAN paper. For example, the batch size is 64 instead of 128, while the original DCGAN paper and its official code neither report results on CIFAR-10 nor Inception scores on any dataset. Besides, the computation of Inception scores in MH-GAN’s official code is different to the [official TF code of Inception scores](https://github.com/openai/improved-gan), particularly in terms of pre-trained Inception models (TF or PyTorch).
>
> Hence, we do not report lower results of existing GANs, but only follow the conventional setup in the literature of sampling methods of GANs. The relatively lower scores may be due to 1) the unstable performance of GANs [1], 2) the difference in hyper-parameters, 3) the difference in Inception score calculation.
>
> Another reason why the literature of sampling methods does not keep up with state-of-the-art results may be that the sampling methods focus on the relative improvement over the base GAN sampling after the training of GANs, and therefore, the absolute value of inception scores may be of less interest. As long as we use the same trained GANs as starting points, only the improvement represents the effectiveness of the sampling methods. According to Table 2, the relative improvement of sampling methods are significant, given that it does not need to re-train the model at all.
>
> We have revised the experiment setup in Section 5.2 and Table 2 caption to make it more clear. We have also stated our implementation on MH-GAN’s official code explicitly in the footnote at page 7.
>
> [1] Lucic et al. Are GANs created equal? A large-scale study. NeurIPS 2018.
>
>
>
> ---
> **Q3**: Minimal improvement of our method over baselines, particularly compared to the wide oscillation of IS over the different epochs.
>
> **A3**: From our view, we think Figure 3 (left) is an important result demonstrating the effectiveness of our method. As shown in Figure 3 (left), the training process of GANs is very unstable, and the inception scores vary a lot from epoch to epoch. To comprehensively evaluate our method, we show the training process of GANs. As shown in Figure 3 (left), our method outperforms previous methods consistently (superior in most epochs) and significantly (improvement > error bar). We have added this discussion in the second paragraph of Section 5.2.
>
> We have also added results on multi-modal distributions in Appendix A.4, where we show our method significantly improves sample quality over baselines while preserving sample diversity in scenarios with increasing modes.
>
> (See more below)

---

### Official Review · AnonReviewer3 · 2020-11-08
**Good direction to move the area of Monte Carlo GAN refinement**

**Rating:** 8
**Confidence:** 3

**Review:**

The paper is a good idea. It is based on the notion of following the same line of reasoning as the prior work of MH-GAN. However, it makes the crucial advance of allowing the use of gradient information for the far more efficient HMC-type samplers.

The paper is able to do much of the sampling in the latent space, which enables the use of gradient-based sampling. One nitpick would be that I suspect there is a latent assumption that the mapping of z -> x is injective (but need not be bijective like in normalizing flows). However, this is not explicitly stated at all in the paper.

---

> ### Author Response · Authors · 2020-11-20
> **Response to Reviewer #3**
>
> Thank you for your positive comments! Here is our answer to your concern.
>
> ---
> Q1: Is there a latent assumption that the mapping of z -> x is injective?
>
> A1: Yes, it is true. When introducing Eq. (8), the change-of-variable formula, we omit technical details to focus on the main story, and instead refer readers to the reference for further details and proof. We have now made these points clear in the newly added Appendix A.1.
>
> First, the mapping should be injective, otherwise there would not be a “one-to-one” change of variables. Second, according to Ben-Israel [1], to ensure Eq. (8) hold, we also need the Jacobian matrix of the mapping to have full column rank for every input. A mild sufficient condition for injectivity is that the generator only contains (non-degenerate) affine layers and injective non-linearities, like LeakyReLU. It is not hard to show that such a condition also implies the full rankness of the Jacobian. In fact, this architecture has already been found to benefit GANs and achieved state-of-the-art results [2]. The affine layers here are also likely to be non-degenerate because their weights are randomly initialized and typically will not degenerate in practice during the training of GANs.
>
> ---
> Thank you again and hope you find our explanations helpful!
>
> [1] Ben-Israel, A. (1999). The change-of-variables formula using matrix volume. SIAM Journal on Matrix Analysis and Applications, 21(1), 300-312.
>
> [2] Tang, S. (2020). Lessons Learned From the Training of GANs on Artificial Datasets. IEEE Access, 8, 165044-165055.

---

### Author Response · Authors · 2020-11-20
**Response to all reviewers**

Thank all reviewers very much for the valuable comments and suggestions.

---
We have made the following updates to address the reviewers’ comments.

+ Title: revised some of words in title from "Coupling Markov chains" to "Reparameterized Markov chains", and the short name, from CMC-GAN to REP-GAN, to avoid confusion in the MCMC community;
+ Section 1: added explanations on why the discriminator can help improve generator sample quality;
+ Section 2: added previous work on combining MCMC and neural networks;
+ Section 4.2: added more explanations to the proof of Theorem 1;
+ Section 4.2: added discussions on our methods as a novel reparameterization technique for general Bayesian models;
+ Section 5.2: added explanations for the left and right panels of Figure 3;
+ Section 5.2: added explanations for the small changes between steps in Figure 4, and added Figure 5 to illustrate the good sample diversity of our proposal with a large step size;
+ Appendix A.1: added a section that explicitly discusses the assumptions of our method;
+ Appendix A.3: added FID results on benchmark datasets, and comparison of computation cost;
+ Appendix A.4: added multi-modal experiments with increasing difficulty.

---
We have revised our paper according to all the valuable comments and please let us know if there is anything still not clear or any other suggestions.

---

### Decision · Program_Chairs · 2021-01-07
**Final Decision**

**Decision:**

Reject

**Comment:**

The consensus among the reviewers is that this is a borderline paper: its main idea is sensible and natural. Unfortunately, while the reviewers appreciated the authors' responses to their comments, they felt that the paper failed to demonstrate the usefulness of the idea beyond toy-datasets. The latter would considerably strengthen this paper.